# Cost, Severity and Prevalence of Agricultural-Related Injury Workers’ Compensation Claims in Farming Operations from 14 U.S. States

**DOI:** 10.3390/ijerph18084309

**Published:** 2021-04-19

**Authors:** Navneet Kaur Baidwan, Marizen R. Ramirez, Fred Gerr, Daniel Boonstra, Joseph E. Cavanaugh, Carri Casteel

**Affiliations:** 1UAB/Lakeshore Collaborative, University of Alabama at Birmingham, Birmingham, AL 35294, USA; 2Division of Environmental Health Sciences, University of Minnesota, Minneapolis, MN 55455, USA; mramirez@umn.edu; 3Occupational and Environmental Health, College of Public Health, University of Iowa, Iowa, IA 52246, USA; fred-gerr@uiowa.edu (F.G.); carri-casteel@uiowa.edu (C.C.); 4Biostatistics, College of Public Health, University of Iowa, Iowa, IA 52242, USA; daniel-boonstra@uiowa.edu (D.B.); joe-cavanaugh@uiowa.edu (J.E.C.)

**Keywords:** agriculture, injuries, workers’ compensation

## Abstract

(1) Background: There is no national surveillance of agricultural injuries, despite agricultural occupations being among the most hazardous in the U.S. This effort uses workers’ compensation (WC) data to estimate the burden of agricultural injuries and the likelihood of experiencing an injury by body part involved, cause, and nature in farming operations. (2) Methods: WC data from 2010 to 2016 provided by a large insurance company covering small to medium-sized farm operations from 14 U.S. states was used. We investigated the associations between injury characteristics and WC costs and the risk of having a more severe versus a less severe claim. The proportion of costs attributable to specific claim types was calculated. (3) Results: Of a total 1000 claims, 67% were medical only. The total cost incurred by WC payable claims (n = 866) was USD 21.5 million. Of this, 96% was attributable to more severe claims resulting in disabilities or death. The most common body part injured was the distal upper extremity. Falling or flying objects and collisions were the most expensive and common causes of injury. (4) Conclusions: Characterizing the cost and severity of agricultural injury by key injury characteristics may be useful when prioritizing prevention efforts in partnership with insurance companies and agricultural operations.

## 1. Introduction

Agricultural work is among the most hazardous occupations in the United States (U.S.) and internationally and affects a large segment of rural populations [1,2,3,4,5,6].

In 2018, workers employed in the U.S. agriculture, forestry, fishing, and hunting (AFFH) sector experienced a fatal injury rate of 23.4/100,000 full-time equivalent (FTE) workers, compared to the overall national average of 3.5/100,000 FTE workers [7]. In addition, the AFFH sector experienced a nonfatal injury and illness rate about two times greater than that of all industries combined [8]. These high rates of injuries are due in part to the variety of hazards to which agricultural workers are exposed [3]. Common hazards include contact with animals [2], trip and fall hazards, and use of tractors and other machinery [4]. Injuries resulting from such hazardous exposures range in severity from minor lacerations and contusions to fatalities [9]. Previously, Leigh, et al. [10] provided national estimates of the costs associated with injuries and illness for agricultural work estimated using the human capital method. The data for these estimates obtained from the Census of Fatal Occupational Injuries, NIOSH National Traumatic Occupational Fatality Survey, and Bureau of Labor Statistics suggest that while agricultural work contributes 1.8% of the GDP, it accounts for 3.5% of the national occupational injury costs, thus warranting greater attention to how these injuries can be prevented. Still, it has now long been acknowledged that lack of knowledge about the magnitude and risk factors for these injuries has remained a barrier for preventive intervention. Despite the magnitude of fatal and nonfatal injury in AFFH, detailed and representative information about agricultural injury events is lacking, which leads to limitations in our ability to accurately characterize the burden of agricultural injuries [1,11]. In the U.S., the absence of federal reporting requirements for injuries and illnesses on farms with fewer than 11 employees limits routine surveillance of agricultural injury by the Bureau of Labor Statistics’ Survey of Occupational Injuries and Illnesses (SOII), which in turn underestimates the magnitude of nonfatal occupational injury and illness to farm workers [12]. Local or state administrative databases are increasingly being used by researchers to characterize agricultural injuries and illness trends [11,13].

While there is no national comprehensive database for occupational injuries and illnesses [14], workers’ compensation (WC) is one of the major sources of such data [15]. WC data cover over 90% of the wage and salary workers in the U.S. Though their primary purpose is to make payments to injured or ill workers [16], WC claims provide data on direct costs and characteristics of injuries across a range of severity, including those that lead to death and disability and those requiring medical treatment only. According to studies examining the overlap between WC and the SOII systems, WC systems are considered more comprehensive and identify more cases of nonfatal occupational injury and illness than the SOII due in large part to the SOII not capturing injuries and illnesses on farms with fewer than 11 employees [17,18]. WC data have been used to describe occupational health outcomes in agriculture, but most of these studies have used data from the state of Washington alone [19,20,21,22]. Furthermore, few studies report the WC costs associated with agricultural injury [21,23]. With the exception of North Dakota, Ohio, Washington, and Wyoming, most U.S. states do not have state-run WC systems.

The overall goal of the current study was to characterize the burden of agricultural injuries first by identifying injury characteristics (i.e., body part affected, cause, and nature of injury) separately for claims requiring medical treatment only and for claims leading to death or disability. We then estimated costs associated with the injury characteristics, stratified by the type of claim (i.e., medical-only and disability/death claims).

## 2. Materials and Methods

A cohort study was conducted of workers’ compensation (WC) claims filed by 662 of 8534 farm policyholders across 14 U.S. states (Arkansas, Georgia, Iowa, Illinois, Indiana, Kansas, Maryland, Michigan, Minnesota, Missouri, Nebraska, Pennsylvania, South Dakota, and Wisconsin) from 2010 to 2016. The farms included in this study were entities involved in the production of commodities (i.e., grain, livestock, and produce) with workers’ compensation coverage by a large underwriter; agribusinesses not involved in commodity production (e.g., grain elevators, feeding mills, packing facilities) were excluded.

Study variables: Claim data from the insurance carrier included the claim type, injury characteristics, claim cost (from wage replacement, medical treatment, and administrative costs such as lawyer fees), and date of injury. Claim type was categorized as medical only, temporary disability, permanent partial and permanent total disability, and death related. For analyses, we categorized claim type into medical-only claims, and disability/death claims.

Injury characteristics available in the claim data were (i) body part injured, (ii) cause of injury, and (iii) nature of injury. Some categories of each of the three injury characteristics were collapsed due to low cell counts. Body part injured was categorized into (i) head, face and neck (ears, eyes, nose, mouth, facial bones, skull, other facial soft tissue), (ii) proximal lower extremity (hip, upper leg, knee), (iii) distal lower extremity (lower leg, ankle, foot, toes), (iv) multiple body parts, (v) torso (abdomen, chest, heart, lungs), (vi) proximal upper extremity (shoulder, upper arm), (vii) distal upper extremity (forearm, elbow, hand, fingers, wrist), (viii) spinal cord and traumatic brain injuries (disc, upper back, lower back, sacrum and coccyx, vertebrae, brain), and (ix) other, unspecified, and internal (soft tissue, unclassified, internal organs). Cause of injury was categorized as (i) animals and insects, (ii) electricity, heat, and temperature extremes, (iii) falling or flying objects, and collisions, (iv) object handling, (v) slips, trips, and falls, (vi) strain (strain or injury by twisting, pushing/pulling, jumping, lifting, reaching, repetitive motion), (vii) tool and machine use, and (viii) other/miscellaneous (not including occupational illness-related causes such as inhalation, absorption, or ingestion of harmful substances). Note that causes of injury categorized as occupational disease or illness were excluded from these analyses. Nature of injury was categorized as (i) amputations and crushing, (ii) contusions, (iii) dislocations and fractures, (iv) sprains and strains, (v) superficial and open wounds, (vi) multiple injuries, and (vii) others (e.g., concussions, poisoning, foreign body, burn, electric shock). Note that natures of injury suggesting an occupational disease or illness with unknown relatedness to injury (e.g., respiratory, mental disorder, cancer, myocardial infarction) were also not included in the analyses.

Statistical analyses: Descriptive statistics (frequency, mean, median, interquartile range, and sum) for costs incurred for claims that resulted in a WC payment were calculated. The total cost incurred was the sum of both the direct payments (wage replacement and medical treatment) and the WC amount reserved to cover administrative costs. Counts and corresponding percentages were calculated for each of the injury characteristics stratified by claim severity.

Generalized linear models (GLMs) based on a marginal binomial distribution and a log link function were used to estimate prevalence ratios of disability/death claims compared to medical-only claims for categories within each of the three injury characteristic groups. The models were fit using generalized estimating equations (GEEs) to account for within-policy correlation. The dependent variable was claim type (disability/death versus medical only). Three separate models were fit using independent variables comprising categories within each of the three injury characteristic groups (i.e., body part injured, cause of injury, and nature of injury), with one category chosen as the referent. Reference categories generally had larger cell sizes than comparison groups and generated effect estimates greater than unity.

Next, GLMs based on a marginal gamma distribution and a log link function were formulated for disability/death and medical-only claims to estimate the mean costs of claims by body part, cause, and nature of injury. Cost models were constructed separately for medical-only and for disability/death claims and for each of the three injury characteristic groups (i.e., six separate models). Again, all models were fit using GEEs to account for the correlation of claims that were from the same policyholder. The overall mean costs were estimated using a weighted average approach, which sums the two model-based means multiplied by the respective relative frequencies of death/disability and medical-only claims. The standard errors for each mean were found by taking the square root of the sum of the squared model-based standard errors multiplied by their respective squared relative frequencies.

We also calculated the proportion of costs attributable to specific types of medical-only claims as a measure of overall burden. The attributable costs of death/disability and medical-only claims were calculated by multiplying the model-based means by the number of claims for each injury characteristic. We then determined the proportion of total costs attributed by each injury characteristic by dividing total costs by the overall cost of the claim types, disability/death, and medical-only claims. We then constructed a bar graph to illustrate the proportion of costs attributable to the key injury characteristics.

## 3. Results

During the study period from 2010 to 2016, a total of 1066 WC claims were processed. Of these claims, 1000 injury claims were analyzed after removing occupational diseases and those with no physical injury. Medical-only claims accounted for 66.6% (n = 666) of the 1000 claims, followed by temporary disability (21.4%), permanent disability (11.5%), and death-related (0.5%) claims.

### 3.1. Body Part Injured, Cause of Injury, and Nature of Injury by Claim Type

Overall, extremity injury was common, with the sum of distal and proximal upper extremity injury accounting for about 37.3% of all claims and the sum of distal and proximal lower extremity injury accounting for about 21.3% of all claims (Table 1). The next most common body parts injured were the head/face/neck (16%) and spinal cord and brain (10.5%). Some differences in claim frequency were observed by claim type (i.e., medical-only versus disability/death claim). Specifically, there were slightly more medical-only claims than disability/death claims for the distal and proximal lower extremity (12.3% vs. 9%) and for the upper extremity (26.2% vs. 11.1%). Head/face/neck, spinal cord and brain, and systemwide and multiple injuries were all more common among disability/death claims than medical-only claims.

When examined by cause of injury, slips, trips, and falls accounted for 19.1% of all claims, followed by animal and insect claims (14.4%), object handling (12.0%), and strain (10.9%). A total of 26% of disability/death claims were caused by slips/trips/falls, 14% by animals/insects, and 13.5% by strain. The frequency of slips/trips/falls, animals/insects and object handling was similar, accounting for 15.8%, 14.6%, and 14.4% of medical-only claims.

The category sprains and strains was the most common nature of injury (26.7%), followed by superficial and open wounds (20.9%). Altogether, sprains/strains (32%) and dislocations/fractures (23%) accounted for more than half of the disability/death claims. Notably, about 30% of the sprains and strains involved the back (results not shown). About half of all medical-only claims were either superficial/open wounds (26.1%) or sprains/strains (24%). Although rare, amputations and crushing injuries were more common among disability/death claims than medical-only claims (5.7% vs. 2%).

Prevalence ratios (PR) of disability/death claims in comparison to medical-only claims by body part injured, cause of injury, and nature of injury are presented in Table 2. The body part with the greatest prevalence of death/disability (in comparison to head/face/neck injury) was the proximal lower extremity (PR = 3.44, 95% CI: 2.26, 5.24). Slips, trips, and falls (PR = 2.21, confidence interval (CI): 1.47, 3.32) and strains (PR = 2.03, CI: 1.33, 3.11) were more than twice as likely as tool/machine use to be causes of disability/death claims. Amputations and crushing injuries (PR = 3.55, CI: 2.39, 5.30) and dislocations and fractures (PR: 3.29, CI: 2.36, 4.60) were more than three times as likely as superficial/open wounds to be a disability/death claim.

### 3.2. Workers’ Compensation Costs of Agricultural Injuries

Of the 1000 claims, 866 (86.6%) resulted in a WC payment, with a total payment of USD 21.5 million over the 7-year study period (Table 3). The mean and median payments per claim were USD 24,829 and USD 1066, respectively. While medical-only claims accounted for 66% (n = 570) of the total claims for which a payment was made, they represented only 4% of WC costs paid by the insurance carrier. In contrast, permanent disability claims were the costliest (more than USD 15 million and 71.1% of all WC costs) but accounted for only 13% of the paid claims. Although death claims were very uncommon (n = 5 death claims), their combined costs were about USD 2.1 million, which represented almost 10% of all WC expenses. The median cost per claim was lowest for medical claims (USD 488) and highest for death-related claims (USD 289,554).

Systemwide injuries and those affecting multiple body parts were the costliest claims (mean = USD 95,238, 95% CI: USD 0, USD 191,485), followed by head, face, and neck and spinal cord and traumatic brain injury claims (Table 4). Overall, falling or flying objects and collisions (mean cost = USD 52,058, CI: USD 4384, USD 99,733) and having multiple injuries (mean cost = USD 136,183, CI: USD 0, USD 348,851) were the most expensive cause and nature of agricultural injury claims, respectively. Further differences were observed for these mean cost estimates by claim severity. For example, systemwide and multiple injuries had the highest mean claim cost leading to death/disability (mean = USD 197,477, CI: USD 71,048, USD 548,888), while torso injuries had highest mean cost of medical-only injury claims (mean = USD 3626, CI: USD 952, USD 13,815).

Finally, we examined the proportion of costs attributable to various claim types. Injuries to head, face, or neck (24.0%) and distal upper extremities (22.5%) accounted for almost half of all costs for medical-only payouts, while systemwide and multiple injuries accounted for 39.3 % of all death/disability claim payouts (Figure 1).

When examining causes of injury, those caused by animals/insects (20.9%) were the most expensive medical-only claims, while slips/trips/falls (16.4%) and falling objects/collisions (19.9%) accounted for over a third of all death/disability claims (Figure 2).

Sprains/strains (25.4%) and superficial/open wounds (21.1%) accounted for about half of all medical-only claim costs. Over 40% of all disability/death payouts were dislocations/fractures (24.1%) and multiple injuries (21.9%) (Figure 3). These injury characteristics are the most expensive because their estimated mean cost and number of incidents were relatively large.

## 4. Discussion

Workers’ compensation (WC) systems are being used increasingly as a surveillance tool to characterize the burden and trends of agricultural injury. In this study, conducted in partnership with a large provider of WC insurance to agricultural operations in the U.S., we evaluated 1000 injury-related workers’ compensation claims processed by the insurer from 2010 to 2016. The total cost of the workers’ compensation claims paid by the insurer during this period of time exceeded USD 21 million. Although two-thirds of claims were categorized as medical-only claims, 91% of the total claims cost was attributed to disability/death claims. Four types of injuries emerged as especially important for prevention efforts because of their frequency and cost: falls, falling/flying objects and collisions, dislocations/fractures, and sprains/strains.

The USD 21 million paid by this insurance carrier for agricultural injury claims is substantial, but within the range of WC costs reported in other studies. A previous study used WC claims between 1996 and 2001 to describe the frequency, severity, and costs associated with injuries experienced by orchard workers in Washington state [20]. Out of 13,068 total claims, 25% were compensable and totaled USD 50.5 million (USD 15,458 mean cost per claim, compared to USD 24,829 mean cost per claim in the current study). Another research effort [24] that analyzed injury claims from low-seam coal mines over a period of 12 years found that WC systems incurred a burden of USD 24 million overall. Out of these, knee, followed by lower back, given their high injury frequencies with a respective burden of around USD 4 million and USD 3 million, cost the most as far as body parts injured are concerned. These were followed by injuries sustained by multiple body parts and systems. The authors report that workers in a coal mine are likely to sustain such injuries since they may be compelled to adopt postures that impose significant load on the musculoskeletal system.

Death/disability claims, which by frequency comprise a third of all claims, were substantially more expensive (median death claims = USD 330,000, median temporary disability claims = USD 19,000, median permanent disability claims = USD 19,000) than medical-only claims (median = USD 488). When prevention resources are limited, intervening on the most debilitating and costly agricultural injuries will likely appeal to operators and insurance carriers alike.

The cause of injury categories of (i) falling or flying objects and collisions, and (ii) slips, trips, and falls accounted for a substantial proportion of severe injury and total workers’ compensation costs. Overall, falling or flying objects and collisions were the most expensive cause of death/disability claims, on average costing about USD 125,000 and representing one in ten claims. Slips, trips, and falls, on the other hand, were twice as frequent, accounting for one in five injuries, but were less costly per claim with an average disability/death claims cost of about USD 40,000. Similar to our study, slips, trips, and falls have been identified as a leading cause of agricultural injury resulting in a WC claim in the U.S. and international settings [1,25,26,27]. However, altogether, injuries from these two causes (i.e., slips, trips, and falls and falling or flying objects and collisions) represented about a quarter of the total costs of death/disability claims in our study.

Dislocations/fractures and sprains/strains were important natures of injury and accounted for almost half of all claims and associated costs. Previous research using WC claims to describe injuries among agricultural workers also reported sprains and strains as the most common nature of injury [20,22,27,28,29]. In our study, about 30% of sprains and strains involved the back, which may account for the relatively high proportion of this nature of injury category that were classified as death/disability claims (data not presented). Our findings are consistent with an older survey of California farm operators [28] and the reported prevalence of strains and sprains reported in the 2019 BLS Survey of Occupational Injuries and Illnesses [30]. These findings underscore the persistent problem of back injuries that often occur among farmers who engage in strenuous lifting, work in awkward positions, and drive farm equipment such as tractors for prolonged periods [27,31].

Partnerships with insurance: Our characterization of agricultural injury patterns was made possible by a unique collaboration between a research university and a major insurance provider that facilitated access to a dataset of claims from 15 states. Such collaboration is particularly fruitful in states lacking a state-run workers’ compensation system. Only Ohio, North Dakota, Wyoming, and Washington have centralized WC programs. Furthermore, our insurance partner provides WC policies for a wide variety of production operations, including those that employ fewer than 11 employees. These smaller types of farms make up a large proportion of farms across the country.

It is possible that WC surveillance studies such as the current study will motivate collaborative and creative prevention efforts among public health researchers, insurance companies and farm operators. For example, insurance companies can serve as important intermediaries for facilitating injury prevention programs on the farm, although intervention research involving insurance companies is still in its nascent stages. For example, one promising study reported that an insurer-supported engineering control program across multiple industries including agriculture, forestry, and fishing/hunting significantly reduced WC claim rates [32]. New areas of translational research are also possible in partnership with insurance companies. For example, engineering and administrative controls, such as those outlined in OSHA standards for walking–working surfaces and fall protection [33], can be adapted for agricultural work. Personal protective equipment may be provided to prevent injuries resulting from falling or flying objects when engineering controls are not available [34]. The translation of these interventions to the agricultural setting can be implemented and evaluated in partnership with insurance companies, which are influential and trusted sources of information to agricultural workers [35].

Use of workers’ compensation and its limitations: WC data are a potential source of information on occupational injuries that may not be available from any other surveillance sources (for example, trauma registries or other hospital records). However, previous studies that used WC data for examining agricultural-related injuries were limited to specific farming practices, such as livestock handling. Additionally, previous research using WC data was state-specific or in international settings, whereas our study analyzed data from multiple states in the U.S. Our study results are limited to the policy holders in this study, and generalization to other populations should be made with caution.

Even though this research provides cost estimates, these costs likely underestimate total injury cost because (i) WC benefits are limited to partial replacement of usual wages and (ii) WC datasets include no information about the indirect costs of injury (e.g., enterprise productivity loss or retraining costs). In addition, the burden of injury and associated costs are likely underestimated from WC records because underreporting of injury and illness to workers’ compensation insurance providers is believed to be common [16,19]. Linking WC data to information obtained by other sources can result in more complete ascertainment of nonfatal injury and illness events [12]. Despite their limitations [11,36,37,38,39], WC data provide useful information regarding the magnitude and characteristics of occupational injury that can be valuable in prioritizing interventions.

## 5. Conclusions

Agricultural workers experience a substantial number of occupational injuries and account for a large financial burden. Surveillance data from WC providers offer a unique opportunity to estimate injury cost and severity, both of which can be important for prioritizing prevention strategies. Slips, trips, and falls as well as flying/falling objects and collisions accounted for the costliest claims and the greatest number of severe claims resulting in death. Prevention efforts that focus on reducing cost, death, and disability simultaneously are likely to be welcome by both agricultural employers and WC insurance providers. With a common goal, partnerships between these agricultural employers and WC providers can lead to the implementation of evidence-based intervention strategies to protect agricultural workers from traumatic injury.

## Figures and Tables

**Figure 1 ijerph-18-04309-f001:**
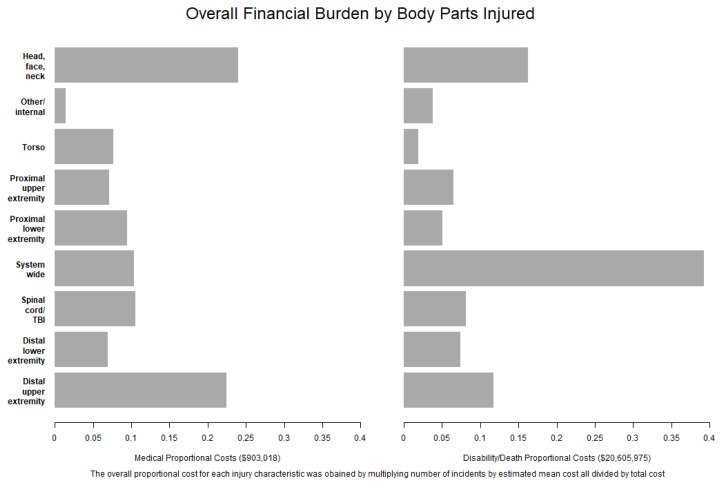
Overall financial burden by the body parts injured.

**Figure 2 ijerph-18-04309-f002:**
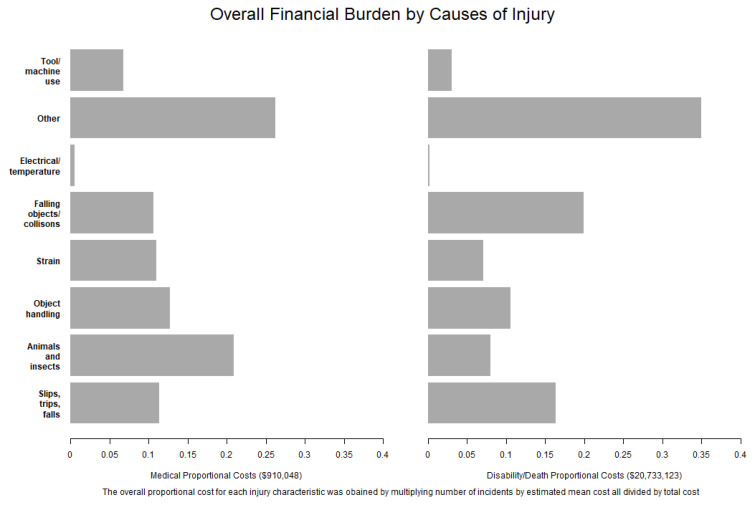
Overall financial burden by the causes of injury.

**Figure 3 ijerph-18-04309-f003:**
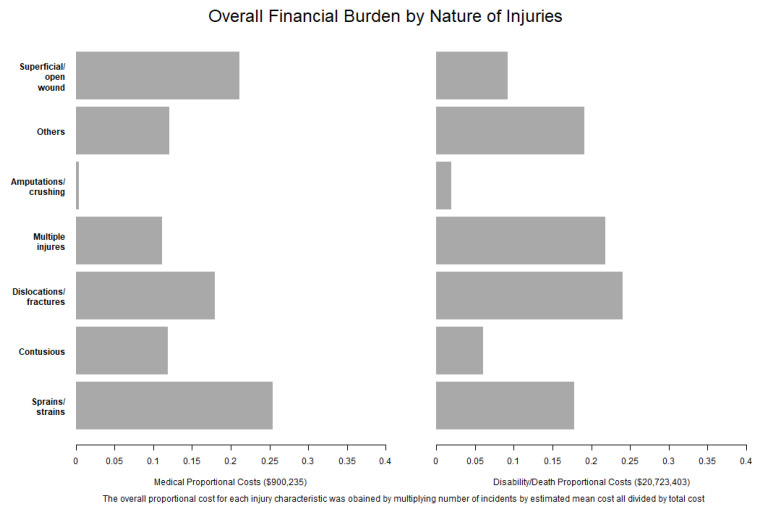
Overall financial burden by the nature of injury.

**Table 1 ijerph-18-04309-t001:** Body part injured, cause of injury, and nature of injury by claim type (medical only and disability/death) (n = 1000).

Injury Characteristic	Claim Type			Total (n = 1000) n (%)
	Medical Only (n = 666) n (%)	Disability/Death (n = 334) n (%)	*p*-Value	
**Body Part Injured**				
Distal upper extremity	213 (32.0)	76 (22.8)	<0.0001	289 (28.9)
Head, face, neck	135 (20.3)	25 (7.5)		160 (16.0)
Distal lower extremity	80 (12.0)	44 (13.2)		124 (12.4)
Spinal cord and traumatic brain injury	62 (9.3)	43 (12.9)		105 (10.5)
Systemwide and multiple injuries	53 (8.0)	47 (14.1)		100 (10.0)
Proximal lower extremity	43 (6.5)	46 (13.8)		89 (8.9)
Proximal upper extremity	49 (7.4)	35 (10.5)		84 (8.4)
Torso	24 (3.6)	11 (3.3)		35 (3.5)
Other, unspecified and internal	7 (1.0)	7 (2.0)		14 (1.4)
**Cause of Injury**				
Slips, trips, and falls	105 (15.8)	86 (25.8)	<0.0001	191 (19.1)
Animals and insects	97 (14.6)	47 (14.1)		144 (14.4)
Object handling	96 (14.4)	24 (7.2)		120 (12.0)
Strain (or injury by twisting, pushing/pulling, jumping, lifting, reaching, repetitive motion)	64 (9.6)	45 (13.5)		109 (10.9)
Tool and machine use	82 (12.3)	20 (6.0)		102 (10.2)
Falling or flying objects, and collisions	54 (8.1)	37 (11.1)		91 (9.1)
Electricity, heat, temperature extremes	8 (1.2)	4 (1.2)		12 (1.2)
Other/Miscellaneous	160 (24.0)	71 (21.3)		231 (23.1)
**Nature of Injury**				
Sprains and strains	160 (24.0)	107 (32.0)	<0.0001	267 (26.7)
Superficial and open wounds	174 (26.1)	35 (10.5)		209 (20.9)
Contusions	113 (17.0)	29 (8.7)		142 (14.2)
Dislocations and fractures	57 (8.6)	76 (22.8)		133 (13.3)
Multiple injuries	24 (3.6)	14 (4.2)		38 (3.8)
Amputations and crushing	13 (2.0)	19 (5.7)		32 (3.2)
Others	125 (18.8)	54 (16.2)		179 (17.9)

**Table 2 ijerph-18-04309-t002:** Prevalence ratios for disability/death claim versus medical-only claim by body part injured, cause of injury, and nature of injury (n = 1000).

Injury Characteristic	* Prevalence Ratios (Disability/Death claims vs. Medical-Only Claims) (95% CI)
**Body Part Injured**	
Distal upper extremity	1.70 (1.13, 2.57)
Distal lower extremity	2.23 (2.26, 5.24)
Spinal cord and traumatic brain injury	2.57 (1.67, 3.94)
Systemwide and multiple injuries	2.92 (1.92, 4.45)
Proximal lower extremity	3.44 (2.26, 5.24)
Proximal upper extremity	2.69 (1.69, 4.27)
Torso	1.96 (1.08, 3.55)
Other, unspecified, and internal	2.98 (1.53, 5.78)
Head, face, and neck	1.00
**Cause of Injury**	
Slips, trips, and falls	2.21 (1.47, 3.32)
Animals and insects	1.63 (1.06, 2.51)
Object handling	1.00 (0.59, 1.68)
Strain (or injury by twisting, pushing/pulling, jumping, lifting, reaching, repetitive motion)	2.03 (1.33, 3.11)
Falling or flying objects, and collisions	1.97 (1.25, 3.09)
Electricity, heat, temperature extremes	1.67 (0.74, 3.76)
Other/Miscellaneous	1.53 (1.00, 2.34)
Tool and machine use	1.00
**Nature of Injury**	
Sprains and strains	2.43 (1.74, 3.40)
Contusions	1.24 (0.82, 1.89)
Dislocations and fractures	3.29 (2.36, 4.60)
Multiple injuries	2.12 (1.24, 3.61)
Amputations and crushing	3.55 (2.39, 5.30)
Others	1.78 (1.22, 2.59)
Superficial and open wounds	1.00

* Log-binomial models accounting for within-policy correlations.

**Table 3 ijerph-18-04309-t003:** Total cost incurred for WC payable claims by claim type (n = 866).

Claim Type		n (%)	Mean (USD)	Median (USD)	IQR * (USD)	Sum (USD) (%)
Medical only		570 (65.8)	1462	488	967	833,094 (3.9)
Disability/death	Temporary Disability	178 (20.6)	18,591	7927	20,054	3,309,218 (15.4)
	Permanent Disability	113 (13.0)	135,210	135,210	84,148	15,278,760 (71.1)
	Death	5 (0.6)	416,182	289,554	289,415	2,080,912 (9.7)
Overall		866	24,829	1066	6757	21,501,985

* Interquartile range.

**Table 4 ijerph-18-04309-t004:** Total cost incurred for WC payable claims by claim type (n = 866).

Injury Characteristic	Mean Workers’ Compensation Cost (USD) * (95% CI)		
	Overall	Medical-Only Claims	Disability/Death Claims
	n = 866	n = 570	n = 296
**Body Part Injured**			
Distal upper extremity	10,221 (6163, 14,280)	1098 (819, 1472)	33,992 (22,117, 52,243)
Distal lower extremity	14,674 (6353, 22,994)	916 (635, 1321)	38,062 (21,098, 68,667)
Spinal cord and traumatic brain injury	20,656 (11,163, 30,149)	1870 (1333, 2623)	48,030 (29,564, 78,030)
Systemwide and multiple injuries	95,238 (0, 191,485)	2087 (1187, 3672)	197,477 (71,048, 548,888)
Proximal lower extremity	14,685 (10,564, 18,807)	2593 (450, 4637)	23,755 (17,601, 32,061)
Proximal upper extremity	20,0076 (13,039, 27,114)	1522 (981, 2363)	47,908 (33,204, 69,124)
Torso	16,153 (0, 36,831)	3626 (952, 13,815)	42,600 (9593, 189,174)
Other, unspecified, and internal	65,939 (0, 145,097)	2129 (1107, 4095)	129,749 (38,302, 439,529)
Head, face, and neck	24,902 (3015, 46,788)	1794 (1003, 3207)	151,995 (59,660, 387,239)
**Cause of Injury**			
Slips, trips, and falls	21,052 (14,066, 28,039)	1174 (906, 1521)	43,479 (30,889, 61,202)
Animals and insects	14,756 (7787, 21,724)	2320 (1322, 4074)	38,469 (22,812, 64,874)
Object handling	22,419 (0, 48,747)	1430 (913, 2238)	99,697 (28,963, 343,178)
Strain (or injury by twisting, pushing/pulling, jumping, lifting, reaching, repetitive motion)	18,201 (10,682, 25,719)	2079 (1523, 2837)	38,565 (24,819, 59,923)
Falling or flying objects, and collisions	52,058 (4384, 99,733)	2003 (1181, 3400)	124,866 (48,920, 318,716)
Electricity, heat, temperature extremes	3647 (1248, 6045)	643 (310, 1334)	10,655 (5067, 22,403)
Other/Miscellaneous	35,970 (0, 73,439)	1634 (921, 2898)	116,825 (39,840, 342,571)
Tool and machine use	7985 (2012, 13,958)	881 (691, 1123)	37,237 (16,391, 84,594)
**Nature of Injury**			
Sprains and strains	17,573 (12,734, 22,411)	1745 (1310, 2325)	40,110 (29,957, 53,703)
Contusions	11,119 (5339, 16,899)	1115 (850, 1462)	48,058 (27,345, 84,461)
Dislocations and fractures	43,617 (10,016, 77,217)	3375 (1754, 6494)	71,211 (32,154, 157,709)
Multiple injuries	136,183 (0, 348,851)	4765 (1508, 15,057)	348,473 (70,645, 1,718,929)
Amputations and crushing	13,148 (6389, 19,907)	312 (216, 450)	21,255 (12,651, 35,710)
Others	26,893 (7111, 46,674)	1008 (77, 1398)	91,904 (43,162, 195,691)
Superficial and open wounds	11,246 (1619, 20,874)	1233 (890, 1709)	57,975 (22,636, 148,487)

* Log-gamma models accounting for within-policy correlations.

## Data Availability

The data for this study cannot be shared.

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
