# Peer review of "Cost, Severity and Prevalence of Agricultural-Related Injury Workers’ Compensation Claims in Farming Operations from 14 U.S. States"

_ijerph, 2021, doi:10.3390/ijerph18084309_

Round 1
Reviewer 1 Report
The theme of paper is interesting, although the authors should make in the introduction a cultural approach to the US health system that would help to understand the idiosyncrasies of insurers and therefore of claims and compensation.
The Introduction is scarce and does not justify the real problem, and bibliographic references are improveable and obsolete.
The objective proposed by the authors is uns ambitious for a longitudinal design of the cohort and the added expenditure that it raises, although the objective described could be the general one, some specifics related to variables as interesting as the provenance of the patients should be included, since it is a study that addresses different STATES.
The most prominent problem with this paper is that the fieldwork of this research was carried out during the years 2010-2016 and we are in 2021, so it should at least justify why the age of this data and take this into account both in the results section and discussion.
The results are shown in an unreactive and non-clarifying manner
The discussion (238-244) refers to another study where the economic differences in compensation are very prominent, but does not explain why of this fact.
In paragraph 252-271, they refer to the consistency of their work, regardless of the age of the data collected that are from 10 years ago.
In paragraph 297-303, it referred to that in its study patients from different US states have taken into account, but it has not been a variable that has been used to associate it with others and estimates more attractive results
Author Response
Thank you for the feedback. Please find our responses to each of the comments below:
The theme of paper is interesting, although the authors should make in the introduction a cultural approach to the US health system that would help to understand the idiosyncrasies of insurers and therefore of claims and compensation.
The Introduction is scarce and does not justify the real problem, and bibliographic references are improveable and obsolete.
R: Thank you for the suggestions. We have now added further details to our introduction section.
The objective proposed by the authors is uns ambitious for a longitudinal design of the cohort and the added expenditure that it raises, although the objective described could be the general one, some specifics related to variables as interesting as the provenance of the patients should be included, since it is a study that addresses different STATES.
R: Thank you for the insights. While we do agree with the author, we did not have variables that could address the stated specifics. For example, while this represents a cohort of policy holders, we have very limited information about the characteristics of the policy holders (per our data use agreement). We do agree that the differences by state might be interesting to explore for future studies, that integrates policy differences by state (this is something we do not have). For the current study, we controlled for state and accounted for within policy correlations. These details have now been added to the footnote for table 4.
The most prominent problem with this paper is that the fieldwork of this research was carried out during the years 2010-2016 and we are in 2021, so it should at least justify why the age of this data and take this into account both in the results section and discussion.
R: We agree that are somewhat dated; however, these data were attained at the start of this CDC-funded study (in 2016). Our data use agreement only allows us data through that year.
The results are shown in an unreactive and non-clarifying manner
R: We acknowledge the reviewer’s comment, but are unable to identify which results should be revised.
The discussion (238-244) refers to another study where the economic differences in compensation are very prominent, but does not explain why of this fact.
R: We have now added details as the author suggested: “Another research effort [20] that analyzed injury claims from low-seam coal mines over a period of 12 years found that WC system incurred a burden of $24 million overall. Out of these, knee, followed by lower back given their high injury frequencies with a respective burden of around $4 million and $3 million cost the most as far as body parts injured are concerned. These were followed by injuries sustained by multiple body parts, and systems. The authors report that workers in a coal mine are likely to sustain such injuries since they may be compelled to adopt postures that impose significant load on the musculoskeletal system.”
In paragraph 252-271, they refer to the consistency of their work, regardless of the age of the data collected that are from 10 years ago.
R: We acknowledge the reviewers' comment. Please refer to our earlier response.
In paragraph 297-303, it referred to that in its study patients from different US states have taken into account, but it has not been a variable that has been used to associate it with others and estimates more attractive results
R: Our analyses are adjusted for “state” and we also account for within policy correlations to address the points that the reviewer rightfully raised. We have now added these details in the footnote for Table 4.
Reviewer 2 Report
This is a well-written and presented study which seeks to catalogue the cost, severity and prevalence of agricultural-related injury workers' compensation claims due to work-related accidents on the farms.
Findings from the study hold salient implications as it provides foundational data which can be deployed towards the development of new strategies and/or platforms for managing or preventing the incidence of the most prevalent types of accidents among the study's population.
The use of data sets from an insurer is noted and the methods utilized for data analysis are also acknowledged. The discussion section is well written and ties to the results of the data sets analyzed.
However, whereas the authors made considerable attempt to post the difference between their study and studies, seeking similar information or contribution, they were silent on the rationale behind the choice of the study area (14 states). Are these the states that have a high prevalence of accidents, or are these states where the operations of the insurer extend to? This justification is imperative as it will contribute towards strengthening the study.
Also, given that the study relied on the data sets from one insurer and incorporated data sets from small farm-holds (with fewer than 11 farmhands), is there a possibility that the authors can compare the findings from this study with the workers' compensation paid by other insurers (a sort of national average) to properly situate the study context within the national WC payment landscape?
Also, it will be interesting to see the prevalence of WC claims in small-farms as compared to larger and/or more mechanized farms within the study context.
Author Response
This is a well-written and presented study which seeks to catalogue the cost, severity and prevalence of agricultural-related injury workers' compensation claims due to work-related accidents on the farms.
Findings from the study hold salient implications as it provides foundational data which can be deployed towards the development of new strategies and/or platforms for managing or preventing the incidence of the most prevalent types of accidents among the study's population.
The use of data sets from an insurer is noted and the methods utilized for data analysis are also acknowledged. The discussion section is well written and ties to the results of the data sets analyzed.
R: We are thankful to the reviewer for their feedback.
However, whereas the authors made considerable attempt to post the difference between their study and studies, seeking similar information or contribution, they were silent on the rationale behind the choice of the study area (14 states). Are these the states that have a high prevalence of accidents, or are these states where the operations of the insurer extend to? This justification is imperative as it will contribute towards strengthening the study.
R: The 14 states included in these analyses are the ones that are covered by the insurance provider that the authors were working with. These details are included in the manuscript as follows: “A cohort study was conducted of worker compensation (WC) claims filed by 662 of 8,534 farm policyholders across 14 US states (Arkansas, Georgia, Iowa, Illinois, Indiana, Kansas, Maryland, Michigan, Minnesota, Missouri, Nebraska, Pennsylvania, South Dakota, and Wisconsin) from 2010 to 2016. The farms included in this study were entities involved in the production of commodities (i.e., grain, livestock and produce) with workers compensation coverage by a large underwriter; agribusinesses not involved in commodity production (e.g., grain elevators, feeding mills, packing facilities) were excluded.”
Also, given that the study relied on the data sets from one insurer and incorporated data sets from small farm-holds (with fewer than 11 farmhands), is there a possibility that the authors can compare the findings from this study with the workers' compensation paid by other insurers (a sort of national average) to properly situate the study context within the national WC payment landscape?
R: Thank you for the suggestion. First, we have now further clarified in our introduction that there does not exist a national database for occupational injuries and illnesses and WC data are one of the major sources of information for such data. Next, as also suggested by another reviewer, we have added the following details about WC costs incurred by another hazardous occupation i.e. coal miners: “Another research effort [20] that analyzed injury claims from low-seam coal mines over a period of 12 years found that WC system incurred a burden of $24 million overall. Out of these, knee, followed by lower back given their high injury frequencies with a respective burden of around $4 million and $3 million cost the most as far as body parts injured are concerned. These were followed by injuries sustained by multiple body parts, and systems. The authors report that workers in a coal mine are likely to sustain such injuries since they may be compelled to adopt postures that impose significant load on the musculoskeletal system.” Our limitations section also provides further that WC data may still underestimate the frequency of such injuries and their overall costs.
Also, it will be interesting to see the prevalence of WC claims in small-farms as compared to larger and/or more mechanized farms within the study context.
R: The authors do raise an interesting question. However, we did not have a farm size indicator variable in our dataset to conduct such analyses. The insurer, for propriety reasons, agreed to provide only lists of policyholders and no other information about these holders.
Reviewer 3 Report
The manuscript presents a study in which the costs associated with injury characteristics are estimated, stratified by the type of claim, and based on data from Workers' compensation (WC) directly obtained through an insurance company. It is an interesting and well-supported work.
However, authors must improve some minor issues before the manuscript can proceed for publication.
- In the Abstract, the objectives must be defined more explicitly.
- "our", "we" frequently occur throughout the text - In scientific language, speech in the first person should not be avoided. Check the entire manuscript.
- Line 171 - "CI" It is not clearly defined.
- Table 3. - "IQR" It is not clearly defined.
- Figure 3.- The scale of the graphs on the XX axis must be the same as that of the previous ones.
- Line 239 - Although the reference to "other studies" is, apparently, associated with the two works mentioned in this same paragraph ([16] e [20]), this is not clear. These works should be mentioned here, as well as other possible studies of the same type.
- Line 245 - The value must be indicated.
Author Response
The manuscript presents a study in which the costs associated with injury characteristics are estimated, stratified by the type of claim, and based on data from Workers' compensation (WC) directly obtained through an insurance company. It is an interesting and well-supported work.
R: Thank you for the positive feedback.
However, authors must improve some minor issues before the manuscript can proceed for publication.
- In the Abstract, the objectives must be defined more explicitly.
R: Thank you we have now edited our abstract as follows: “(1) Background: Representative information about agricultural injuries, despite it being among the most hazardous occupations in the U.S., is lacking.”
- "our", "we" frequently occur throughout the text - In scientific language, speech in the first person should not be avoided. Check the entire manuscript.
R: While we agree that third person is used in traditional scientific literature, more contemporary studies now include first person. This approach also leads to more active rather passive voice Line 171 - "CI" It is not clearly defined.
R: We have now defined CI when it is used first (Line 171).
- Table 3. - "IQR" It is not clearly defined.
R: We have now defined IQR (Line 195)
- Figure 3.- The scale of the graphs on the XX axis must be the same as that of the previous ones.
R: Thank you. The scales have now been made consistent among the figures.
- Line 239 - Although the reference to "other studies" is, apparently, associated with the two works mentioned in this same paragraph ([16] e [20]), this is not clear. These works should be mentioned here, as well as other possible studies of the same type.
R: Thank you. The following details have now been added: “Another research effort [20] that analyzed injury claims from low-seam coal mines over a period of 12 years found that WC system incurred a burden of $24 million overall. Out of these, knee, followed by lower back given their high injury frequencies with a respective burden of around $4 million and $3 million cost the most as far as body parts injured are concerned. These were followed by injuries sustained by multiple body parts, and systems. The authors report that workers in a coal mine are likely to sustain such injuries since they may be compelled to adopt postures that impose significant load on the musculoskeletal system.”
- Line 245 - The value must be indicated.
R: The overall cost ($24 million) and other specifications have now been added as discussed above.